# Revealing intrinsic domains and fluctuations of moiré magnetism by a wide-field quantum microscope

Mengqi Huang[1,2,9], Zeliang Sun[3,9], Gerald Yan[1], Hongchao Xie [3], Nishkarsh Agarwal [4], Gaihua Ye[5], Suk Hyun Sung [4], Hanyi Lu[1], Jingcheng Zhou [1,2], Shaohua Yan[6], Shangjie Tian [6,7], Hechang Lei [6], Robert Hovden [4], Rui He [5], Hailong Wang[2,8], Liuyan Zhao [3] ✉ & Chunhui Rita Du [1,2,8] ✉

Moiré magnetism featured by stacking engineered atomic registry and lattice interactions has recently emerged as an appealing quantum state of matter at the forefront of condensed matter physics research. Nanoscale imaging of moiré magnets is highly desirable and serves as a prerequisite to investigate a broad range of intriguing physics underlying the interplay between topology, electronic correlations, and unconventional nanomagnetism. Here we report spin defect-based wide-field imaging of magnetic domains and spin fluctuations in twisted double trilayer (tDT) chromium triiodide $CrI_3$. We explicitly show that intrinsic moiré domains of opposite magnetizations appear over arrays of moiré supercells in low-twist-angle tDT $CrI_3$. In contrast, spin fluctuations measured in tDT $CrI_3$ manifest little spatial variations on the same mesoscopic length scale due to the dominant driving force of intralayer exchange interaction. Our results enrich the current understanding of exotic magnetic phases sustained by moiré magnetism and highlight the opportunities provided by quantum spin sensors in probing microscopic spin related phenomena on two-dimensional flatland.

Moiré magnetism, an emergent class of quantum states of matter, features periodically modulated magnetic order and interaction, and provides an appealing platform for exploring emergent spin transport and dynamic behaviors in solid states[1–10]. Very recently, the microscopic spin arrangement within individual magnetic moiré supercells has been directly visualized in twisted double trilayer (tDT) chromium triiodide $CrI_3$[2]. Same as all spontaneous symmetry breaking phases, degenerate domain states of moiré magnetism are anticipated and

should manifest on length scales across multiple moiré wavelengths. However, there have been few studies on the intrinsic domain phases and structures of moiré magnetic orders. Here we report nitrogen-vacancy (NV) center-based[11–13] wide-field imaging of magnetic domains and spin fluctuations in tDT $CrI_3$. We show that intrinsic moiré domains of opposite magnetizations appear on a mesoscopic length scale in low-twist-angle (0.3°) tDT $CrI_3$, and that the formed domain states can be trained by applying a small external magnetic field. In contrast, such

¹Department of Physics, University of California, San Diego, La Jolla, CA 92093, USA. ²School of Physics, Georgia Institute of Technology, Atlanta, GA 30332, USA. ³Department of Physics, University of Michigan, Ann Arbor, MI 48109, USA. ⁴Department of Material Science and Engineering, University of Michigan, Ann Arbor, MI 48109, USA. ⁵Department of Electrical and Computer Engineering, Texas Tech University, Lubbock, TX 79409, USA. ⁶Laboratory for Neutron Scattering, and Beijing Key Laboratory of Optoelectronic Functional Materials MicroNano Devices, Department of Physics, Renmin University of China, Beijing 100872, China. ⁷School of Materials Science and Engineering, Anhui University, Hefei 230601, China. ⁸Center for Memory and Recording Research, University of California, San Diego, La Jolla, CA 92093, USA. ⁹These authors contributed equally: Mengqi Huang, Zeliang Sun. ✉e-mail: lyzhao@umich.edu; c1du@physics.ucsd.edu

mesoscopic domain features are absent in large-twist-angle (15°) tDT and pristine $CrI_3$. Our work adds a further ingredient for the burgeoning topic of moiré magnetism, highlighting the significant potential of quantum metrology in studying unconventional nanomagnetism hosted by exotic condensed matter systems.

Moiré superlattices consisting of atomically thin van der Waals crystals have attracted tremendous attention on the forefront of quantum materials research study[14,15]. By stacking layers of two-dimensional (2D) materials with a small twist angle or a lattice mismatch, a plethora of exotic electronic, photonic, and magnetic phases can be created and engineered due to the introduction of a periodically modulated atomic registry on the scale of the moiré wavelengths. Notable examples include flat band-based correlated and topological electronic states[14,15], moiré magnetism[1–10], and moiré excitons[16–19]. Over the past few years, 2D materials such as graphene[14,15,20,21], transition metal dichalcogenides[16,18,19,22], and van der Waals magnets[1–5] have been under intensive investigations in this context, and transformative quantum technologies built on moiré materials are underway.

In contrast to the study of the charge degree of freedom which has achieved a remarkable success in controlling the electronic and excitonic properties of moiré quantum matter[14,15], the magnetic counterpart, moiré magnetism, remains relatively underexplored. An apparent challenge results from the limited experimental tools capable of resolving spatially varying magnetic patterns hosted by moiré materials at the nanoscale. While the noncollinear spin textures[5,10], topological skyrmion lattices[6,7], stacking dependent magnetism[8] and magnon bands[9] have been theoretically predicted in van der Waals magnet-based moiré superlattices, real-space imaging of these emergent magnetic features remains as a formidable challenge at the current state of the art. Here we explore NV centers[11–13], optically active atomic defects in diamond, to perform wide-field magnetometry imaging[23–26] of twisted $CrI_3$. Taking advantage of the appreciable field sensitivity and spatial resolution of NV centers, we have observed stacking-induced intrinsic (ferro)magnetic domains spontaneously formed over arrays of moiré supercells in low-twist-angle tDT $CrI_3$. Furthermore, we reveal a uniform spatial distribution of spin fluctuations in tDT $CrI_3$ despite the presence of magnetic domains, suggesting that spin fluctuations in moiré magnets are mainly driven by the intralayer exchange interaction instead of the spatially modulated interlayer coupling.

## Results

Before discussing the details of our experimental results, we first present the device structures and our measurement platform as illustrated in Fig. 1a. In this work, we fabricated tDT $CrI_3$ devices by the standard "tear-and-stack" technique and encapsulated them with hexagonal boron nitride (hBN) nanoflakes[1–3,20,27]. Figure 1b shows a prepared tDT $CrI_3$ sample (top) made from an atomically thin $CrI_3$ flake (bottom). The twist angle was controlled to be -0.3° in order to obtain a relatively large moiré period of -130 nm. It is worth mentioning that the microscopic lattice structure of prepared low-twist-angle tDT $CrI_3$ typically shows certain distortions from the expected hexagonal superlattice pattern, which could be induced by lattice strain, relaxation, and local inhomogeneities[28] (see Supplementary Information Note 1 for details). The prepared device was released onto a diamond membrane for NV wide-field magnetometry measurements. NV centers at the diamond surface are created by $^{14}N^+$ ion implantation with an energy of 3 keV, and the depth of implanted NVs is estimated to be -10 nm[29]. In the current study, we utilize NV ensembles to perform nanoscale imaging of the static magnetic textures and dynamic spin fluctuations in prepared tDT $CrI_3$ devices. Magnetic circular dichroism (MCD) measurements[5] on control samples are used to qualitatively diagnose the magnetic properties of twisted and pristine $CrI_3$.

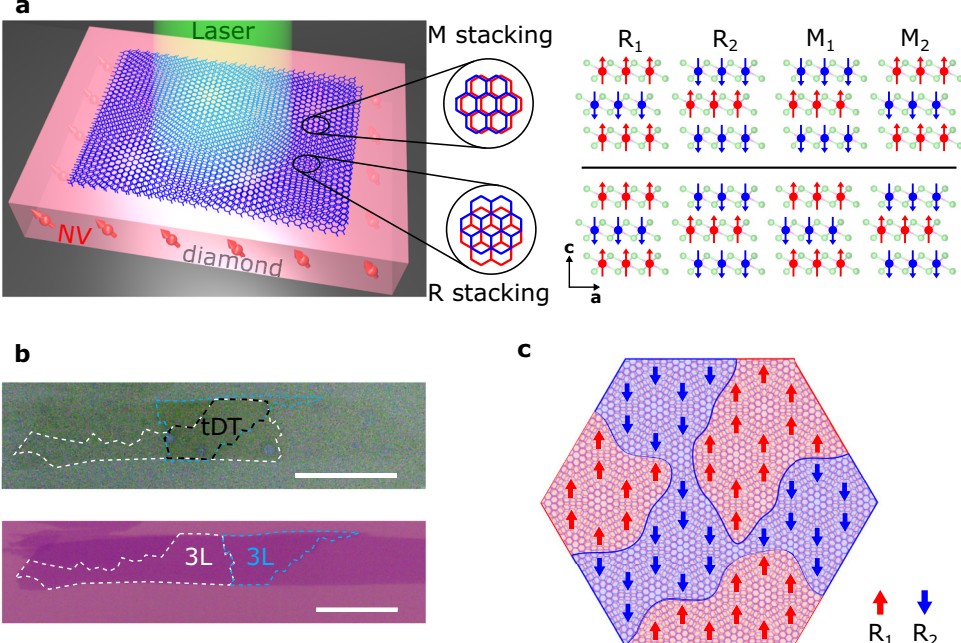

**Fig. 1 | Device structure and NV measurement platform. a** Left: Schematic illustration of NV wide-field magnetometry measurements of twisted $CrI_3$. Right: Monoclinic (AB') and rhombohedral (AB) stacking-induced two-fold degenerate antiferromagnetic ($M_1$ and $M_2$) and ferromagnetic ($R_1$ and $R_2$) phases in the ground state of tDT $CrI_3$. The red and blue arrows represent local magnetization carried by Cr atoms (red and blue balls) at individual layers. The light green balls represent the I atoms. **b** Optical microscopy image of a tDT $CrI_3$ sample on a diamond membrane (top) prepared from a large-sized trilayer $CrI_3$ flake on a Si/$SiO_2$ substrate (bottom) by the "tear and stack" technique. The twisted area is outlined by the black dashed lines, and the original trilayer $CrI_3$ flake and the tearing boundary is marked with the white and blue dashed lines. Scale bar is 20 μm. **c** Schematic of the twisted interface of tDT $CrI_3$ with a low twist angle. Rhombohedral (AB) stacking-induced ferromagnetic order shows $R_1$ (red) and $R_2$ (blue) domains with opposite polarity on an extended length scale that is larger than the moiré period. The red and blue arrows denoting the spin-up and spin-down along the out-of-plane directions, respectively, are used to illustrate the ferromagnetic order at local rhombohedral stacking sites.

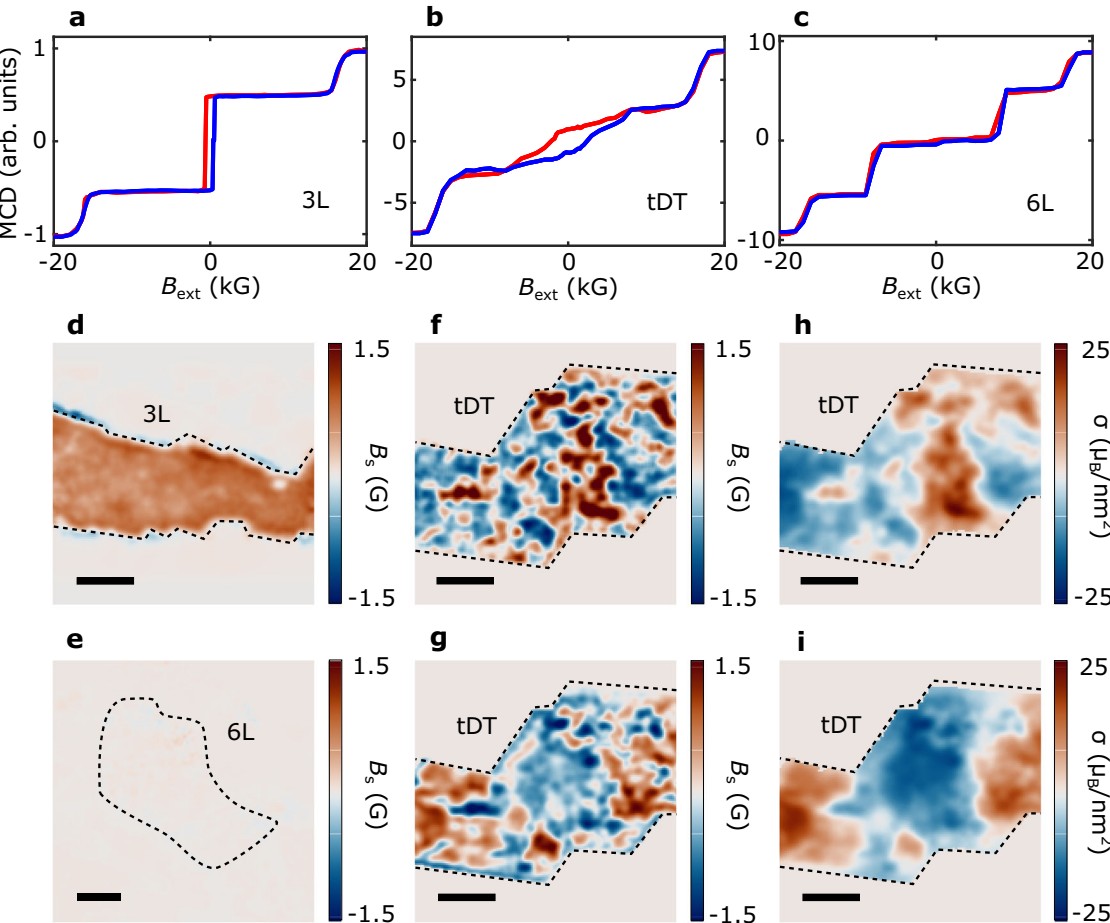

**Fig. 2 | MCD and NV wide-field measurements of CrI₃ samples. a–c** MCD signals as a function of an out-of-plane magnetic field for trilayer, 0.3° tDT, and six-layer CrI₃ samples. The blue and red curves correspond to increasing and decreasing magnetic field, respectively. **d, e** NV wide-field imaging of magnetic stray fields emanating from pristine trilayer (**d**) and six-layer (**e**) CrI₃ samples. **f, g** Magnetic stray field patterns measured for the 0.3° tDT CrI₃ sample with a positive (**f**) and negative (**g**) cooling field. **h, i** Reconstructed magnetization maps of the tDT CrI₃ sample measured with a positive (**h**) and negative (**i**) cooling field. The black dashed lines outline the boundary of the CrI₃ samples of interest, and the scale bar is 3 μm for all the images.

Pristine CrI₃ belongs to the A-type antiferromagnet family showing the characteristic even-odd layer-number determined (un)compensated magnetization in the magnetic ground state[30,31]. For twisted CrI₃, the magnetic moment is spatially modulated within a moiré unit cell depending on the local atomic registry[1–3]. The magnitude and sign of the interlayer exchange coupling varies from the monoclinic (AB') to rhombohedral (AB) stacking geometries leading to co-existing ferromagnetic and antiferromagnetic order in tDT CrI₃, as illustrated in Fig. 1a. Such stacking induced spin arrangement within individual moiré supercells has been visualized by scanning NV magnetometry in a previous work[2]. The current study mainly focuses on real-space imaging of the energetically degenerate domain states, with opposite magnetizations and related by the time-reversal operation, formed over arrays of moiré supercells in tDT CrI₃. Such domain states should extend over multiple moiré periods at a mesoscopic length scale as illustrated in Fig. 1c, and their degeneracy is controllable by external stimuli such as magnetic field, thermal cycles, and local defects. The prominent spatial magnetic "inhomogeneity" together with reduced inter-domain coupling naturally results in spontaneous formation of stacking-induced (ferro)magnetic domains in tDT CrI₃. A major goal of the present work is to utilize NV centers, a sensitive probe to local stray fields, to identify the real-space distribution of these novel magnetic domain states.

We first use MCD to qualitatively reveal the magnetic properties of pristine and twisted CrI₃ samples. The MCD measurement is

sensitive to the total magnetization perpendicular to the sample surface, serving as an ideal experimental probe for investigating the magnetic ground state of CrI₃ with out-of-plane anisotropy[1,3,5]. Our MCD measurements were performed at a temperature of 12 K with an out-of-plane magnetic field, and the measured magneto-optical signals were averaged over a micrometer-sized laser spot on the sample surface. Figure 2a–c show field dependent MCD signals of pristine trilayer, 0.3° tDT, and six-layer CrI₃ samples, respectively. The trilayer CrI₃ with uncompensated magnetization in the ground state exhibits a characteristic hysteresis loop centered at zero magnetic field and spin-flip transitions at ± 1.6 tesla (T). And the six-layer CrI₃ sample with fully compensated net magnetization shows vanishingly small MCD signals at zero magnetic field and two field-driven magnetic phase transitions at ± 0.7 T and ± 1.6 T. Notably, 0.3° tDT CrI₃ exhibits a mixture of the magnetic phases of the trilayer and six-layer samples, suggesting co-existence of ferromagnetic and antiferromagnetic order due to the spatially modulated stacking geometries.

Next, we present NV wide-field magnetometry[23–26] results to image the real space magnetic patterns of the CrI₃ devices. Wide-field magnetometry exploits the Zeeman effect[12] of NV ensembles to measure local magnetic stray fields emanating from the CrI₃ devices (see Supplementary Information Note 2 for details). Figure 2d and e show 2D magnetic stray field maps of pristine trilayer and six-layer CrI₃ samples measured at a temperature $T = 5$ K with an out-of-plane magnetic field $B_{ext} = 71$ G. The trilayer CrI₃ sample exhibits a robust stray field $B_s$

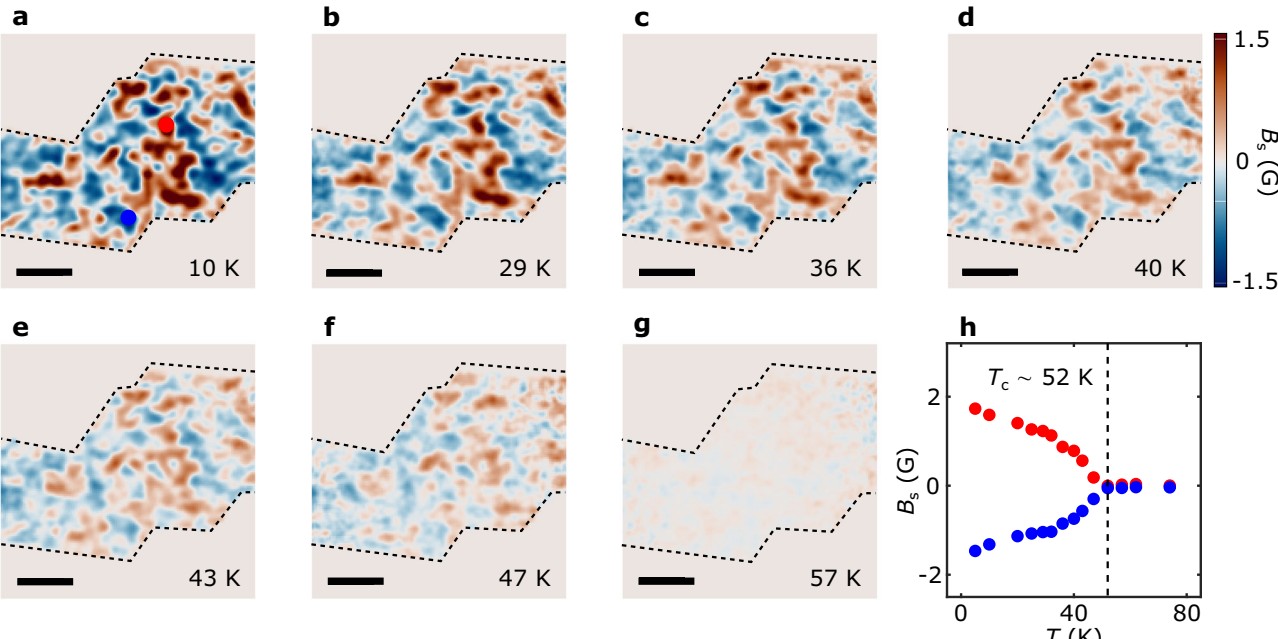

**Fig. 3 | Temperature dependence of twisted magnetism. a–g** Stray field patterns of the 0.3° tDT CrI$_3$ sample measured at an out-of-plane field $B_{ext}$ = 71 G and temperatures of 10 K (**a**), 29 K (**b**), 36 K (**c**), 40 K (**d**), 43 K (**e**), 47 K (**f**), and 57 K (**g**), respectively. The black dashed lines outline the boundary of the 0.3° tDT CrI$_3$, and the scale bar is 3 μm. **h** Temperature dependence of the emanating stray field measured at two local magnetic domain sites with opposite polarity, from which the Curie temperature of the 0.3° tDT CrI$_3$ is estimated to be 52 K (black dashed lines).

arising from the uncompensated magnetic moment while the six-layer one shows a vanishingly small stray field as expected due to its antiferromagnetic ground state. In contrast with the nearly uniform stray field distribution in the pristine samples, the tDT CrI$_3$ device, after field cooling with a positive field of ~71 G, shows distinct multidomain features with stray fields of opposite polarity emanating from individual domains (Fig. 2f). When reversing the sign of the external magnetic cooling field, the polarity of the individual domains also changes (Fig. 2g), indicating that the stacking-induced magnetic degeneracy can be controlled by the thermal cycle and field history. Through well-established reverse-propagation protocols, the corresponding magnetization maps of the tDT CrI$_3$ sample can be reconstructed (Figs. 2h,i, see Supplementary Information Note 3 for details). One can see that the tDT CrI$_3$ sample consists of magnetic domains of opposite magnetizations (~ ±15 μ$_B$/nm$^2$). The nonvanishing magnetic moment results from the one layer of uncompensated magnetization for each individual CrI$_3$ trilayer as illustrated in Fig. 1a. It is worth mentioning that the measured magnetization reflects a spatial average over the ferromagnetic and antiferromagnetic regions within moiré supercells, from which the fraction of rhombohedral order is estimated to be 60% in 0.3° tDT CrI$_3$. The lateral dimensions of the observed magnetic domains in tDT CrI$_3$ lies on the micrometer length scale, which is orders of magnitude larger than the estimated moiré wavelength. The opposite signs of the measured static magnetizations in combination with the mesoscopic characteristic length scale suggest that the observed magnetic patterns result from intrinsic (ferro)magnetic domains consisting of multiple moiré supercells in tDT CrI$_3$ (see Supplementary Information Note 4 for details). The spatial distribution of the extended magnetic domains in tDT CrI$_3$ is co-determined by the intrinsic material properties including competition between dipole and exchange interactions, magnetic anisotropy, local defects, strain, as well as the external experimental stimuli. It is instructive to note that such multidomain features disappear in tDT CrI$_3$ devices with a large twist angle (15°), where pure ferromagnetic order emerges uniformly, showing a clear single magnetic domain (see Supplementary Information Note 5 for details).

We now present systematic NV wide-field magnetometry results to reveal the magnetic phase transition of the tDT CrI$_3$ sample across the Curie temperature. Figure 3a–g show the magnetic stray field maps of the prepared 0.3° tDT CrI$_3$ device measured at temperatures varying from 10 K to 57 K with an external magnetic field $B_{ext}$ of 71 G. In general, the magnetic stray field emanating from the tDT CrI$_3$ sample decreases with increasing temperature due to reduced static magnetization. In the low-temperature regime ($T \leq 36$ K), tDT CrI$_3$ exhibits robust magnetization owing to the suppressed thermal fluctuations as shown in Fig. 3a–c. When approaching the magnetic phase transition temperature, the magnetization dramatically decays accompanied by blurring of the magnetic domain boundaries in tDT CrI$_3$ (Fig. 3d–f). Above the Curie temperature, the magnetization distribution pattern gradually disappears over the entire device area (Fig. 3g). Figure 3h summarizes temperature dependent evolution of the magnetic stray field $B_s$ measured at two local domain sites with opposite polarities. The emanating magnetic field exhibits a gradual decay in the low-temperature regime ($T < 30$ K), followed by a dramatic drop during the magnetic phase transition of tDT CrI$_3$ (see Supplementary Information Note 3 for details).

In addition to the d.c. magnetometry measurements presented above, NV centers, known as spin qubits with excellent quantum coherence, provide additional opportunities for probing non-coherent fluctuating magnetic fields that are challenging to access by the conventional magnetometry methods[32–37]. Next, we apply NV spin relaxometry techniques to spatially image magnetic fluctuations in the prepared tDT CrI$_3$ device. Thermally induced spin fluctuations in a magnetic sample couple to proximal NV centers through the dipole-dipole interaction. Fluctuating magnetic fields at the NV electron spin resonance (ESR) frequencies induce NV spin transitions from the m$_s$ = 0 to m$_s$ = ±1 states, leading to enhancement of the corresponding NV spin relaxation rates[26,35,36,38]. By measuring the spin-dependent NV photoluminescence, the occupation probabilities of NV spin states can be quantitatively obtained, allowing for extraction of the NV spin relaxation rate which is proportional to the magnitude of the local fluctuating

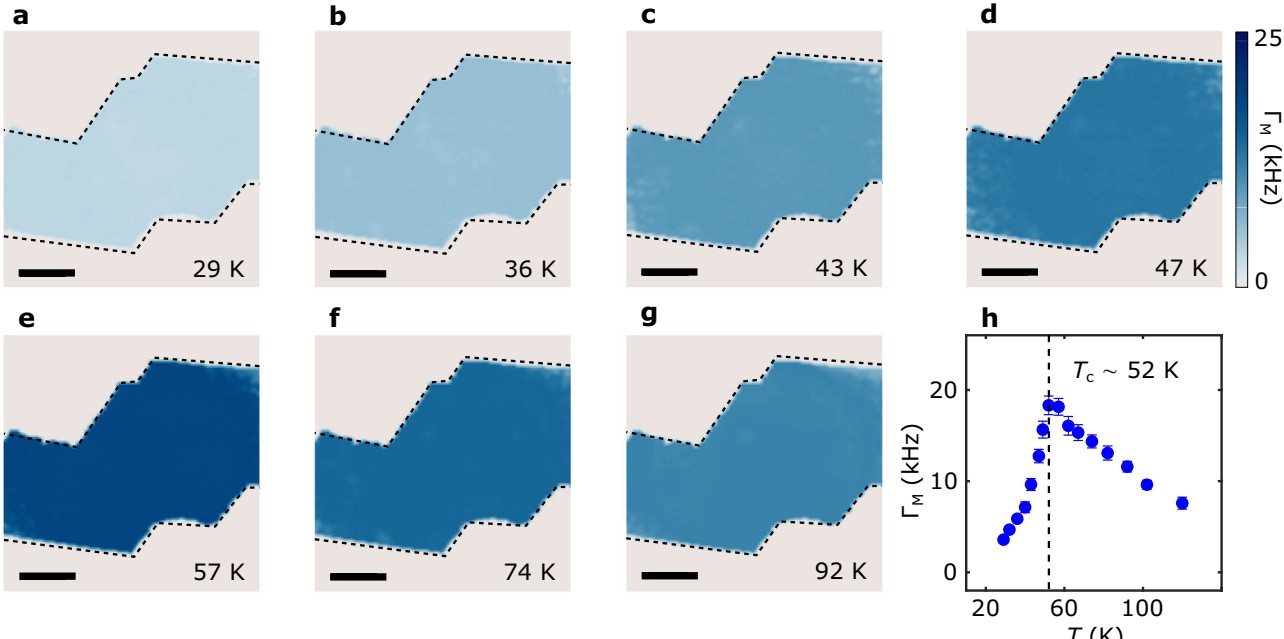

**Fig. 4 | Temperature dependence of spin fluctuations in twisted CrI₃. a–g** NV spin relaxation maps for the 0.3° tDT CrI₃ sample measured at temperatures of 29 K (**a**), 36 K (**b**), 43 K (**c**), 47 K (**d**), 57 K (**e**), 74 K (**f**), and 92 K (**g**), respectively. The NV ESR frequency is set to be 2.78 GHz in these measurements with an external out-of-plane magnetic field $B_{ext}$ = 71 G. The black dashed lines outline the boundary of the tDT CrI₃ sample, and the scale bar is 3 μm. **h** Temperature dependence of NV spin relaxation rate $\Gamma_M$ measured at NV centers underneath the tDT CrI₃ device. The dashed lines mark the estimated Curie temperature of tDT CrI₃.

magnetic field transverse to the NV axis[38,39] (see Supplementary Information Note 6 for details).

Figure 4a–g present a series of NV spin relaxation rate maps of the 0.3° tDT CrI₃ device measured in a broad temperature range. The background of intrinsic NV spin relaxation has been subtracted to highlight the contribution from the magnetic sample. In the current study, the minimum magnon energy of CrI₃ is larger than the NV ESR frequencies under our experimental conditions. Thus, the measured NV spin relaxation is mainly driven by the longitudinal spin fluctuations of CrI₃, which are further related to the static longitudinal magnetic susceptibility and the diffusive spin transport constant[26,39]. In the low-temperature regime ($T < 40$ K), magnetic fluctuations in CrI₃ are largely suppressed due to the vanishingly small spin susceptibility, resulting in reduced NV spin relaxation rate (Fig. 4a and b). As the temperature increases, the measured NV spin relaxation rate significantly increases near the magnetic phase transition of tDT CrI₃ and reaches the maximum value around the critical point (Fig. 4c–e), which is attributed to the dramatic enhancement of spin susceptibility of tDT CrI₃ around the Curie temperature. When temperature is above the magnetic phase transition point, spin fluctuations remain active in tDT CrI₃ due to the finite spin-spin correlation in the paramagnetic state[26,40] (Fig. 4f–g). Figure 4h summarize the temperature dependence of the measured NV spin relaxation rate $\Gamma_M$ with a peak value of 18 kHz around the Curie temperature.

Interestingly, the measured spin fluctuations exhibit a largely uniform spatial distribution over the tDT CrI₃ sample area, in sharp contrast with the static magnetic stray field patterns showing the distinct multidomain features. The magnitude of the spin fluctuations is fundamentally correlated with the spin diffusion constant $D$ and longitudinal magnetic susceptibility $\chi_0$ governed by the exchange energy of magnetic lattices. While the stacking geometries spatially modulate the interlayer coupling strength of the tDT CrI₃ sample, the dominant intralayer exchange interaction largely remains the same. The minimal spatial variation of the measured NV spin relaxation rate indicates that spin fluctuations in tDT CrI₃ are mainly driven by the intralayer exchange interaction while the role of the interlayer

coupling is secondary. Invoking a theoretical model developed in ref. 26, the longitudinal magnetic susceptibility $\chi_0$ and spin diffusion constant $D$ of the tDT CrI₃ sample is extracted to be $(4.0 \pm 0.2) \times 10^{-2}$ emu · cm⁻³ · Oe⁻¹ and $(4.2 \pm 0.3) \times 10^{-5}$ m²/s at 40 K from the NV relaxometry results (see Supplementary Information Note 7 for details). The spin diffusion constant $D$ reflects the intrinsic spin transport capability of a magnetic system, which is further related to other important material parameters such as spin decay (diffusion) length[39]. $D$ is fundamentally determined by the magnon velocity $v$ and the momentum relaxation time $\tau$ as follows: $D = \frac{v^2 \tau}{2}$ [26]. Using the obtained spin diffusion constant value, the magnon velocity $v$ in tDT CrI₃ is estimated to be ~4.3 km/s when taking a momentum scattering time $\tau \sim 5$ ps, which is in qualitative agreement with the theoretical estimation[7,41]. The extracted longitudinal magnetic susceptibility $\chi_0$ describes dynamic magnetic responses along the magnetic order direction of tDT CrI₃. It is typically anticipated that $\chi_0$ shows a divergent behavior across the second order phase transition. A detailed knowledge of this material parameter of atomically thin van der Waals magnets, as demonstrated in the current study, will provide an alternative way to investigate the local magnetic phase variations of emergent material systems on 2D flatland. Building on the current study, we further share the optimism that it would be very interesting to explore the relative contributions of interlayer and intralayer exchange coupling driven spin fluctuations in twisted CrI₃ with different layer thicknesses. Meanwhile, local variations of spin fluctuations may also emerge within individual moiré supercells between the magnetized rhombohedral stacking and zero magnetization monoclinic stacking sites. Here we reserve these exciting experiments for a future study where more advanced NV microscopy techniques with enhanced spatial resolution may be used.

## Discussion

In summary, we have demonstrated NV wide-field imaging of the magnetic domains formed at a mesoscopic length scale of multiple moiré periods in tDT CrI₃. By using NV spin relaxometry techniques, we further probe the spin fluctuations in twisted CrI₃, whose magnitude

reaches a maximum around the magnetic phase transition point. In contrast with the static magnetic stray field patterns showing distinct multidomain features, spin fluctuations driven by the intralayer exchange interaction exhibits a largely uniform spatial distribution in tDT $CrI_3$. We note that multiple tDT $CrI_3$ samples have been evaluated to ensure the consistency of the presented results (see Supplementary Information Note 8 for details). Our work highlights the significant potential of NV centers for investigating the local static and dynamic magnetic behaviors in emergent moiré superlattices, suggesting new opportunities for probing the interplay between "inhomogeneous" magnetic order, spin transport and dynamic behaviors in a broad range of quantum states of matter.

## Methods
### Materials and device fabrications
$CrI_3$ crystals used in this study were grown by the chemical vapor transport method as reported in a previous literature[3]. Atomically thin $CrI_3$ flakes were first exfoliated onto $Si/SiO_2$ substrates. The layer number was determined by thickness-dependent optical contrast and further confirmed by MCD measurements. We made tDT $CrI_3$ devices by using a polymer-stamping technique and encapsulated them by hBN nanoflakes. We first used poly(bisphenol A carbonate) stamp to pick up a top hBN and tear a selected trilayer $CrI_3$ flake into two parts. One piece was picked up by the top hBN on the stamp, and the other one remained on the $Si/SiO_2$ substrate was rotated by a well-controlled angle. The two $CrI_3$ flakes were stacked with each other to form a twisted device and finally encapsulated by a bottom hBN flake. The entire device fabrication processes involving handling $CrI_3$ flakes were performed inside a nitrogen-filled glovebox with water and oxygen levels below 0.1 ppm. We prepared a total of four tDT $CrI_3$ devices on diamond membranes for NV magnetometry measurements. The experimental NV data measured on device A are presented in the main text, and the NV results of devices B, C and D are included in the Supplementary Information. Note that samples with other layer thicknesses such as the small-twist-angle double bilayer $CrI_3$ and twisted bilayer $CrI_3$ are not studied here due to the lack of net ferromagnetism or potential device quality issues.

### NV magnetometry measurements
Pulsed NV ESR and spin relaxometry measurements were performed using a wide-field microscope. The prepared $CrI_3$ samples were positioned in a closed-cycle optical cryostat allowing for measurements from 4.5 K to 350 K. Microsecond-long green laser pulses used for NV spin initialization and readout were generated by an electrically driven 515 nm laser. The laser beam spot width after passing the objective was about 20 μm × 20 μm, and was subsequently focused on the diamond surface. NV fluorescence was imaged using a CMOS camera. Pulses to drive the green laser and to trigger the camera exposure were generated by a programmable pulse generator. Continuous microwave currents were generated using Rohde & Schwarz SGS100a and/or Rohde & Schwarz SMB100a signal generators. Nanosecond-long microwave current pulses were generated by sending continuous microwave currents to a microwave switch (Minicircuits ZASWA-2-50DR+) electrically controlled by a programmable pulse generator. The microwave pulses were sent through a microwave combiner (Mini-Circuits ZB3PD-63-S+) and amplified by +50 dB (Mini-Circuits ZHL-25W-63+) before being delivered to the on-chip Au stripline patterned on diamond samples. The external magnetic field applied in our NV measurements was generated by a cylindrical NdFeB permanent magnet attached to a scanning stage inside the optical cryostat. Further details of the measurement protocol for NV ESR and spin relaxometry are discussed in Supplementary Information Notes 2 and 6. The single crystal diamond membranes containing shallowly implanted NV centers used in this work are commercially available from the company Qnami.

### MCD measurements
MCD measurements were performed at 12 K by using a 632.81 nm laser. The laser beam was focused on the sample with a spot size of ~2–3 μm. The polarization of the incident light was modulated between right and left circular polarization by using a photoelastic modulator (Hinds Instruments PEM-200) and the MCD signal was measured by demodulating the reflectivity signal against the frequency of the photoelastic modulator.

## Data availability
All data supporting the findings of this study are available from the corresponding author on reasonable request.

## Code availability
All code not included in the paper are available upon reasonable request from the corresponding authors.

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

## Acknowledgements

Authors thank Raymond Zhu for insightful discussions. M. H., H. L., H. W., and C. R. D. were supported by the Air Force Office of Scientific Research under award No. FA9550–20-1-0319 and its Young Investigator Program under award No. FA9550-21-1-0125. G. Q. Y., J. Z., and C. R. D. acknowledged the support from U. S. National Science Foundation (NSF) under award No. DMR-2046227 and ECCS-2029558. L. Z. acknowledges support by NSF CAREER grant no. DMR-174774, AFOSR YIP grant no. FA9550-21-1-0065, and Alfred P. Sloan Foundation. R. He acknowledges support by NSF grant no. DMR-2104036. R. Hovden acknowledges support from the DOE Office of Basic Energy Sciences. N.A. and S.H.S. acknowledge support from the US Army Research Office (W911NF-22-1-0056). H. C. L was supported by the National Key R&D Program of China (Grant No. 2018YFE0202600, 2022YFA1403800), the Beijing Natural Science Foundation (Grant No. Z200005), and the National Natural Science Foundation of China (12274459).

## Author contributions

M. H. performed the NV measurements and analyzed the data with G. Q. Y., H. L., J. Z., and H. W. Z. S., H. X. and L. Z. fabricated the $CrI_3$ devices. G. Y. and R. He performed the MCD measurements. N. A., S. H. S., and R. Hovden performed the transmission electron microscopy characterizations. S. Y., S. T., and H. C. L. provided bulk $CrI_3$ crystals. C. R. D. supervised this project.

## Competing interests

The authors declare no competing interests.
