## [Peer Review File · Nature Communications]

Reviewers' Comments:

Reviewer #1:

Remarks to the Author:

The authors report the NV measurement of a magnetic moire system, which is a very popular topic recently. Thanks to the field sensitivity and spatial resolution of the experimental technique, twisted double trilayer CrI₃ was studied by the wide-field magnetometry. Magnetic domains and their magnetization were clearly imaged. In addition, spin fluctuations were also measured for the first time in this system. In my opinion, the experimental data is nicely presented with good interpretations. Here are some of my questions and concerns:

(1) The magnetism in twisted CrI₃ has been predicted and carefully studied by the same group, for example, in references 3 and 5. The imaging of these magnetic domains is expected and won't add too much information for further understanding the moire magnetism here.

(2) The main result, in my opinion, is the spin fluctuations measurement, which could be very interesting due to these magnetic domains formed at the moire length scale. However, it seems only uniform spatial distribution of the spin fluctuations was observed, due to the dominant effect from the intralayer exchange interactions. This is also expected and explained by the authors.

(3) The authors didn't explain why only twisted double trilayer samples were studied here, not the twisted double bilayer in their previous studies, and also not other layer thicknesses.

(4) Following (2), since it is a problem that the intralayer exchange interactions are stronger than the interlayer coupling, one may expect a relatively stronger interlayer effect when the layer thickness is reduced here?

With these being said, especially given its broad Nat. Comm. readership, I cannot recommend it for publication, at least in the present form.

Reviewer #2:

Remarks to the Author:

In their manuscript, "Revealing intrinsic domains and fluctuations of moire magnetism by a wide-field quantum microscope," Huang et al. experimentally investigate the magnetism of twisted double-trilayer chromium tri-iodide (tDT CrI₃) using magnetic circular dichroism and wide-field nitrogen-vacancy (NV) magnetometry and relaxometry. They uncover micron-scale ferromagnetic domains which have spatial extent larger than the moire period for the twist-angle used, and additionally uncover that the direction of the magnetization of these domains depends on the direction of an externally applied magnetic field during sample cooling below the Curie temperature. They propose a model for the existence of such extended ferromagnetic domains. Their wide-field NV measurements of the magnetization, determined by the stray field detected by a proximal ensemble of NV centers, reveals the spatial orientation and strength of the magnetization and demonstrate the vanishing magnetization at the Curie temperature. NV relaxometry measurements probing the longitudinal spin fluctuations in the tDT CrI₃ reveal a spin noise spectrum which does not vary spatially, suggesting that interlayer physics dominates the spin fluctuations. Last, they apply an established model of the NV center relaxation rate to determine the longitudinal magnetic susceptibility and spin diffusion constant of their sample. The authors' results are an interesting and exciting extension of previous NV magnetometry studies of such 2D magnetic systems.

The impact of these results would benefit from a clearer presentation. I hope that the following remarks and questions are of assistance in guiding the manuscript towards that clarity:

1. The central piece of this study, from which all other results derive, is the existence of magnetic domains on length scales that exceed the moire period. The importance of this observation and the proposed model warrants a clearer discussion.

a. What generates the spatial extent of such domains?

b. Are the directions of the red and blue arrows in Figure 1(c) uniquely determined by the moire pattern underneath, or are the directions of each red and blue arrow assigned schematically, meant to help the reader understand that extended ferromagnetic islands can be supported?

c. Is the proposed model of extended ferromagnetic domains in tDT CrI₃ (say, Supplementary

Figure 3) original to this manuscript or is there some existing or related study that could be cited?
d. It would be useful if the authors could provide comment on the differences between their observations of extended domains in tDT CrI3 and the results reported in the previous NV magnetometry study of similar tDT CrI3 samples (the 2021 study in Science), where such extended domains were not observed.

2. A few remarks on the NV relaxometry results.

a. The NV relaxometry results versus temperature for temperatures above the Curie temperature in Figure 4(h) are surprising. One might expect that above the Curie temperature the disappearance of the magnetization would cause the NV relaxation rate to restore to near the background rate of about 4 kHz, as indicated in the supplement. Why does the NV relaxation rate continue to be enhanced up to and above 100K?

b. The authors could provide better context for the final relaxometry results. The observation of no spatial dependence of the relaxation rate, the conclusion about inter-layer effects dominating the spin noise spectrum, and the concluding calculations of the longitudinal magnetic susceptibility and the spin diffusion constant are great results. What's not clear from the manuscript as written is what a general audience should be taking away from these findings. It would be helpful to provide a bit of discussion about what these numbers mean.

3. Circular dichroism measurements

a. Did the authors perform spatially-resolved circular dichroism measurements on the tDT sample? The resolution of that technique (1 μm) is not totally dissimilar to the wide-field imaging NV technique (500 nm), and would be interesting to compare, if possible. Knowing if the circular dichroism technique could be applied to study similar samples may be of benefit to a general audience since it can be less experimentally demanding than NV imaging. Last, a circular dichroism image could be useful for direct comparison to the reconstructed magnetization maps from the NV measurements.

-Brendan McCullian

Reviewer #3:

Remarks to the Author:

The manuscript by Mengqi Huang et al. has experimentally studied the twisted double trilayer CrI3 with the spin defect-based wide-field imaging technique. They were able to observe magnetic domains and reveal spin fluctuations in the sample. Similar technique has been utilized by the same group in investigating other 2D magnetic materials such as MnBi4Te7 and Fe3GeTe2. A slightly different but closely related scanning NV magnetometry measurements have previously been carried out on twisted bilayer CrI3 and double-trilayer CrI3 in imaging the magnetic domains (ref.2). I don't see this study has significantly advanced the understanding of the moiré magnetism physics or any technique improvement that warrant publication in Nature Communications.

Meanwhile, the authors take it for granted that structurally the twisted region of their sample is uniform. However, as generally known that, for small-angle twisted (0.3 degree) sample studied here, the superlattices can hardly be uniform in mesoscopic length scales. It's highly possible that the twisted region of the sample is already relaxed into random large domains with different and complicated interlayer stackings, which give rise to the domain structures observed by the wide-field imaging.

Response to reviews – NCOMMS-22-52510-T

We thank the reviewers for their detailed and constructive assessments of our manuscript. Based on their responses, we have implemented all of the suggestions that they have kindly provided, and as a result have strengthened the conclusion of our manuscript and made it more accessible to the general audiences. In particular, we have extended our discussions to further present the significance and potential impact of the spin fluctuation study of moiré magnetism in the current work. Below, we address the reviews on a point-by-point basis.

Reviewer #1:

The authors report the NV measurement of a magnetic moire system, which is a very popular topic recently. Thanks to the field sensitivity and spatial resolution of the experimental technique, twisted double trilayer CrI₃ was studied by the wide-field magnetometry. Magnetic domains and their magnetization were clearly imaged. In addition, spin fluctuations were also measured for the first time in this system. In my opinion, the experimental data is nicely presented with good interpretations. Here are some of my questions and concerns:

Response: We thank the reviewer for his/her endorsement of our work. The reviewer's technical questions will be addressed in detail below.

(1) The magnetism in twisted CrI₃ has been predicted and carefully studied by the same group, for example, in references 3 and 5. The imaging of these magnetic domains is expected and won't add too much information for further understanding the moire magnetism here.

Response: We would like to respectfully point out that the current work on moiré magnetism study is clearly distinct from previous works in the following aspects:

New physics: The current study, for the first time, explicitly shows the extended domain phases across multiple moiré wavelengths in tDT CrI₃ as illustrated in Fig. R1. In contrast, the previous scanning NV work [Ref. 2, *Science* **374**, 1140 (2021)] mainly investigates the microscopic spin arrangement within individual magnetic moiré supercells. Also, our work reports the first study of nanoscale spin fluctuations in moiré magnetic superlattices using NV relaxometry methods.

New device platform: the presented work focuses on twisted double trilayer (tDT) CrI₃ device systems, while Refs. 3 and 5, as mentioned by the reviewer, study twisted double bilayer (tDB) CrI₃.

New experimental techniques: the current work utilizes state-of-the-art quantum imaging techniques to spatially visualize mesoscopically extended magnetic domain distributions in tDT CrI₃, allowing us to probe new science that is challenging to access by conventional magneto-optical methods.

Figure R1. Schematic of rhombohedral (AB) stacking-induced extended ferromagnetic domains R₁ (red) and R₂ (blue) across multiple moiré periods in tDT CrI₃. The red and blue arrows denoting the spin-up and spin down along the out-of-plane directions, respectively, are used to illustrate the ferromagnetic order at local rhombohedral stacking sites.

In summary, the current work builds on new material/device platforms and experimental tools to deliver previously inaccessible information on moiré magnetism, such as extended magnetic domains, local spin fluctuations, longitudinal magnetic susceptibility, and spin diffusion constant, bringing new insights to this highly popular and fast-growing research field.

(2) The main result, in my opinion, is the spin fluctuations measurement, which could be very interesting due to these magnetic domains formed at the moire length scale. However, it seems only uniform spatial distribution of the spin fluctuations was observed, due to the dominant effect from the intralayer exchange interactions. This is also expected and explained by the authors.

Response: We thank the reviewer for his/her recognition of our results. The reviewer correctly points out that spin fluctuations in extended magnetic domains are driven by the dominant intralayer exchange interaction, manifesting little variations for the time-reversal related domain states in twisted CrI₃. Inspired by his/her comments, we realize that it would be very interesting to study variations of local spin fluctuations across magnetic domain walls formed in moiré magnetism. We also anticipate variations of spin fluctuations at different local stacking sites within individual moiré supercells, i.e., magnetized R stacking and zero magnetization M stacking sites. However, the widths of these “intercell” domain walls and lateral dimensions of stacking induced “intracell” magnetic patches may be in the tens of nanometers regime, which is beyond the spatial sensitivity of our wide-field NV imaging techniques. So, we reserve these exciting experiments for future studies where cutting-edge scanning NV microscopy with enhanced spatial resolution will be used.

In summary, we hope that the spin fluctuation measurements of tDT CrI₃ in the current work, whose results are expected and perhaps less surprising, serves the purpose of stimulating further exciting studies on this topic. In this revision, we have added an outlook on this point in the first paragraph on page 6 as follows:

“.....Building on the current study, we further share the optimism that it would be very interesting to explore the relative contributions of interlayer and intralayer exchange coupling driven spin fluctuations in twisted CrI₃ with different layer thicknesses. Meanwhile, local variations of spin fluctuations may also emerge within individual moiré supercells between the magnetized rhombohedral stacking and zero magnetization monoclinic stacking sites. Here we reserve these exciting experiments for a future study where more advanced NV microscopy techniques with enhanced spatial resolution may be used.”

(3) The authors didn't explain why only twisted double trilayer samples were studied here, not the twisted double bilayer in their previous studies, and also not other layer thicknesses.

Response: We thank the reviewer for the detailed point. This work focuses on twisted double trilayer CrI₃ to investigate extended magnetic domains formed in moiré magnetism, which remains unexplored in the current state of the art. We also used NV wide field techniques to investigate local magnetic textures of twisted double bilayer CrI₃. In the small twist angle regime (~0.3 degree), we did not observe robust stacking induced (ferro)magnetism which is expected due to the fully compensated net magnetization (in agreement with Dr. Zhao's work: Refs. 3 and 5). Other layer thicknesses such as twisted bilayer CrI₃ devices may have additional complications brought by the

reduced crystalline and magnetic quality of monolayer and twisted bilayer CrI₃ as reported by previous work [Ref. 2 and *PNAS* **117**, 24664 (2020)]. In this revision, we have further clarified this point at the end of the Materials and device fabrications section as follows:

“.....Note that samples with other layer thicknesses such as the small-angle twisted double bilayer CrI₃ and twisted bilayer CrI₃ are not studied here due to the lack of net ferromagnetism or potential device quality issues.”

(4) Following (2), since it is a problem that the intralayer exchange interactions are stronger than the interlayer coupling, one may expect a relatively stronger interlayer effect when the layer thickness is reduced here?

Response: The reviewer raises an excellent question on the thickness dependent interlayer interaction in twisted CrI₃. We agree with the reviewer that such an effect could potentially be more pronounced in the reduced sample thickness regime. As discussed in our response above, twisted double bilayer and twisted bilayer CrI₃ devices are not studied here due to the lack of net ferromagnetism and potential device quality issues, respectively. So unfortunately, we cannot experimentally investigate this interesting topic in the current work. In this revision, we have added an outlook on this point in the first paragraph on page 6 (see our response to reviewer’s point 2).

This also comes back to our response above: scanning NV microscopy with an enhanced spatial resolution will be an ideal measurement tool to study the local variations of the interlayer effect within moiré supercells. Currently, we are working towards this direction, and hope that our future work will provide more insights on this topic.

With these being said, especially given its broad Nat. Comm. readership, I cannot recommend it for publication, at least in the present form.

Response: We again thank the reviewer for the detailed and constructive assessment of our work. We have provided the requested information to sufficiently address the reviewer’s technical questions and comments. We believe that our work will generate interest amongst many branches on the forefront of quantum sensing and condensed matter physics research, and thus will be of great interest to the broad readership of *Nature Communications*.

As pointed out by the reviewer, moiré magnetism is a “very popular topic recently”, and we hope that our results will contribute to the development of this rapidly growing and advancing field.

Reviewer #2:

In their manuscript, “Revealing intrinsic domains and fluctuations of moire magnetism by a wide-field quantum microscope,” Huang et al. experimentally investigate the magnetism of twisted double-trilayer chromium tri-iodide (tDT CrI₃) using magnetic circular dichroism and wide-field nitrogen-vacancy (NV) magnetometry and relaxometry. They uncover micron-scale ferromagnetic domains which have spatial extent larger than the moire period for the twist-angle used, and additionally uncover that the direction of the magnetization of these domains depends on the direction of an externally applied magnetic field during sample cooling below the Curie temperature. They propose a model for the existence of such extended ferromagnetic domains.

Their wide-field NV measurements of the magnetization, determined by the stray field detected by a proximal ensemble of NV centers, reveals the spatial orientation and strength of the magnetization and demonstrate the vanishing magnetization at the Curie temperature. NV relaxometry measurements probing the longitudinal spin fluctuations in the tDT CrI₃ reveal a spin noise spectrum which does not vary spatially, suggesting that interlayer physics dominates the spin fluctuations. Last, they apply an established model of the NV center relaxation rate to determine the longitudinal magnetic susceptibility and spin diffusion constant of their sample. The authors' results are an interesting and exciting extension of previous NV magnetometry studies of such 2D magnetic systems.

Response: We thank the reviewer for the positive assessment of our manuscript.

The impact of these results would benefit from a clearer presentation. I hope that the following remarks and questions are of assistance in guiding the manuscript towards that clarity:

1. The central piece of this study, from which all other results derive, is the existence of magnetic domains on length scales that exceed the moiré period. The importance of this observation and the proposed model warrants a clearer discussion.

a. What generates the spatial extent of such domains?

Response: We thank the reviewer for the constructive point. The spatial distribution of the magnetic domain states extending over multiple moiré periods in tDT CrI₃ is co-determined by the intrinsic material properties including competition between dipole and exchange interactions, magnetic anisotropy, local defects, strain, as well as external experimental stimuli. In this work, we have shown that the cooling field (Figs. 2f-2i) and temperature (Fig. 3) could effectively control and modulate the spatial distribution of domain phases formed in tDT CrI₃. This is a highly interesting research topic that only emerged very recently. A comprehensive understanding of the fundamental properties of moiré magnetic domains calls for future experimental and theoretical work. And we hope that the current study will stimulate more research interest and efforts in this direction. In this revision, we have further refined the relevant discussions on this point in the first paragraph on page 4 as follows:

“.....The spatial distribution of the extended magnetic domains in tDT CrI₃ is co-determined by the intrinsic material properties including competition between dipole and exchange interactions, magnetic anisotropy, local defects, strain, as well as the external experimental stimuli.....”

b. Are the directions of the red and blue arrows in Figure 1(c) uniquely determined by the moiré pattern underneath, or are the directions of each red and blue arrow assigned schematically, meant to help the reader understand that extended ferromagnetic islands can be supported?

Response: We thank the reviewer for the careful review. The red and blue arrows in Fig. 1(c) are mainly used to help readers understand that magnetic domain phases are formed across multiple moiré periods in twisted CrI₃. In this revision, we have modified Fig. 1(c) and further clarified this point in the figure caption to avoid any potential misunderstanding:

“.....Rhombohedral (AB) stacking-induced ferromagnetic order shows R₁ (red) and R₂ (blue) domains with opposite polarity on an extended length scale that is larger than the moiré period. The red and blue arrows denoting the spin-up and spin-down along the out-of-plane directions, respectively, are used to illustrate the ferromagnetic order at local rhombohedral stacking sites.”

c. Is the proposed model of extended ferromagnetic domains in tDT CrI₃ (say, Supplementary Figure 3) original to this manuscript or is there some existing or related study that could be cited?

Response: The characteristic hexagonal magnetic lattice structure (Supplementary Fig. 3a) of individual moiré unit cells was originally proposed in Ref. 2 [*Science* **374**, 1140 (2021)]. To simulate the mesoscopic stray field patterns of extended magnetic domains in tDT CrI₃, here we randomly distribute the hexagonal moiré cells with positive (red) and negative (blue) ferromagnetic order over a lateral sample area of $\sim 2 \mu\text{m} \times 2 \mu\text{m}$ (Supplementary Fig. 3b). The simulation results show individual domains with opposite polarities on a micrometer length scale (Supplementary Fig. 3c), in qualitative agreement with our experimental results. In this revision, we have further clarified these technical details in Supplementary Note 3 as follows:

“In this section, we qualitatively simulate the stray field patterns arising from the stacking induced (ferro)magnetic domains formed over arrays of moiré supercells in the ground state of low-twist-angle tDT CrI₃. Using the theoretical model proposed in Ref. 6, Supplementary Fig. 3a shows the characteristic hexagonal magnetic lattice structure of two moiré unit cells of tDT CrI₃ with opposite ferromagnetic order.....”

d. It would be useful if the authors could provide comment on the differences between their observations of extended domains in tDT CrI₃ and the results reported in the previous NV magnetometry study of similar tDT CrI₃ samples (the 2021 study in Science), where such extended domains were not observed.

Response: The reviewer raises an excellent point on why the extended domain features in tDT CrI₃ are not observed in the previous work [Ref. 2, *Science* **374**, 1140 (2021)]. For this point, we believe that the different experimental measurement length scales and magnetic field conditions are the two major reasons.

First, Ref. 2 [*Science* **374**, 1140 (2021)] mainly focuses on the local spin arrangement of twisted CrI₃ within individual moiré supercells on a spatial length scale of tens of nanometers. In contrast, the current work utilizes wide-field techniques capable of visualizing magnetic structures of twisted CrI₃ over a much larger spatial range of $\sim 20 \mu\text{m} \times 20 \mu\text{m}$. The “broad” field of view enabled by our techniques provides an excellent opportunity for probing the extended moiré domain phases across multiple moiré periods that has not been revealed in the previous studies.

Secondly, the NV wide-field imaging results reported in the current work were measured under a small external magnetic field of ~ 70 G. It is instructive to note that the previous scanning NV studies [*Science* **374**, 1140 (2021)] were performed under a much higher magnetic field of ~ 2000 G with large field cooling (~ 5000 G). Application of a large external (cooling) field will likely result in a uniform magnetic phase, which does not favor the formation of extended domains

in tDT CrI₃. We again thank for the reviewer for the constructive point. In this revision, we have highlighted these technical details at the end of Supplementary Note 3 as follows:

“Here, we would like to highlight that the “broad” field of view (~20 μm × 20 μm) enabled by our NV wide field techniques provides an excellent opportunity for investigating the extended moiré domain phases across multiple moiré periods in tDT CrI₃, which has not been revealed in the previous study⁶. It is also instructive to note that our measurements were mostly performed under a small external magnetic field (tens of Gauss), which is favorable to the formation of spatially varying domain states in tDT CrI₃ especially in the magnetic phase transition regime.”

2. A few remarks on the NV relaxometry results.

a. The NV relaxometry results versus temperature for temperatures above the Curie temperature in Figure 4(h) are surprising. One might expect that above the Curie temperature the disappearance of the magnetization would cause the NV relaxation rate to restore to near the background rate of about 4 kHz, as indicated in the supplement. Why does the NV relaxation rate continue to be enhanced up to and above 100K?

Response: We thank the reviewer for the detailed point. The enhanced NV spin relaxation rate observed during the magnetic phase transition of tDT CrI₃ (Fig. 4h) is mainly driven by its divergent magnetic susceptibility around the Curie temperature (T_c). When temperature is below T_c , magnetic fluctuations in tDT CrI₃ are largely suppressed due to the vanishingly small magnetic susceptibility. When temperature is above T_c , spin fluctuations remain active in tDT CrI₃ due to the finite spin-spin correlation in the paramagnetic state. It is worth mentioning that similar phenomena were also observed in other van der Waals magnets such as MnBi₄Te₇ [*Nano Letters* **22**, 5810 (2022)] and Fe₃GeTe₂ [*Nature Communications* **13**, 5369 (2022)] in previous NV studies. In this revision, we have added the relevant discussion in the second paragraph on page 5.

“.....When temperature is above the magnetic phase transition point, spin fluctuations remain active in tDT CrI₃ due to the finite spin-spin correlation in the paramagnetic state^{26,39} (Figs. 4f-4g).”

b. The authors could provide better context for the final relaxometry results. The observation of no spatial dependence of the relaxation rate, the conclusion about inter-layer effects dominating the spin noise spectrum, and the concluding calculations of the longitudinal magnetic susceptibility and the spin diffusion constant are great results. What’s not clear from the manuscript as written is what a general audience should be taking away from these findings. It would be helpful to provide a bit of discussion about what these numbers mean.

Response: We thank the reviewer for the constructive point to improve the presentation of our manuscript. In this revision, we have added further discussion on the longitudinal magnetic susceptibility and spin diffusion constant extracted from NV relaxometry results in the third paragraph on page 5 as follows:

“.....The spin diffusion constant D reflects the intrinsic spin transport capability of a magnetic system, which is further related to other important material parameters such as spin decay (diffusion) length³⁸. D is fundamentally determined by the magnon velocity v and the momentum relaxation time τ as follows: $D = \frac{v^2\tau}{2}$ ²⁶. Using the obtained spin diffusion constant value, the magnon velocity v in tDT CrI₃ is estimated to be ~ 4.3 km/s when taking a momentum scattering time $\tau \sim 5$ ps, which is in qualitative agreement with the theoretical estimation^{7,40}. The extracted longitudinal magnetic susceptibility χ_0 describes dynamic magnetic responses along the magnetic order direction of tDT CrI₃. It is typically anticipated that χ_0 shows a divergent behavior across the second order phase transition. A detailed knowledge of this material parameter of atomically thin van der Waals magnets, as demonstrated in the current study, will provide an alternative way to investigate the local magnetic phase variations of emergent material systems on 2D flatland.”

3. Circular dichroism measurements

a. Did the authors perform spatially-resolved circular dichroism measurements on the tDT sample? The resolution of that technique (1 μ m) is not totally dissimilar to the wide-field imaging NV technique (500 nm), and would be interesting to compare, if possible. Knowing if the circular dichroism technique could be applied to study similar samples may be of benefit to a general audience since it can be less experimentally demanding than NV imaging. Last, a circular dichroism image could be useful for direct comparison to the reconstructed magnetization maps from the NV measurements.

Response: We thank the reviewer for the excellent point. For our current MCD apparatus, the position of the beam spot could only be manually controlled over the sample surface and lacks any automated scanning capabilities. Thus unfortunately, we cannot achieve the proposed high-resolution MCD mapping measurements. We agree with the reviewer that fruitful information could be extracted from direct comparison of the magnetization maps measured by these two complementary techniques (NV and MCD imaging); we shall leave this work to our future endeavors.

Reviewer #3:

The manuscript by Mengqi Huang et al. has experimentally studied the twisted double trilayer CrI₃ with the spin defect-based wide-field imaging technique. They were able to observe magnetic domains and reveal spin fluctuations in the sample. Similar technique has been utilized by the same group in investigating other 2D magnetic materials such as MnBi₄Te₇ and Fe₃GeTe₂. A slightly different but closely related scanning NV magnetometry measurements have previously been carried out on twisted bilayer CrI₃ and double-trilayer CrI₃ in imaging the magnetic domains (ref.2). I don't see this study has significantly advanced the understanding of the moiré magnetism physics or any technique improvement that warrant publication in Nature Communications.

Response: We respectfully disagree with the reviewer's evaluation of the significance and impact of the current study. We believe that our work is clearly distinct from the previous works including *Science* **374**, 1140 (2021) in the following two aspects:

New physics: Ref. 2 [*Science* **374**, 1140 (2021)] mainly reports stacking induced ferromagnetic and antiferromagnetic order within individual moiré supercells. In the current work, for the first time, we explicitly show that extended intrinsic moiré magnetic domains of opposite magnetizations appear over arrays of moiré supercells in low-twist-angle tDT CrI₃, and that the formed domain states can be effectively trained by a small external magnetic field. Taking advantage of NV relaxometry methods, we further probe intrinsic spin fluctuations in tDT CrI₃, whose magnitude reaches a maximum around the magnetic phase transition point. All these new insights on moiré magnetism were not revealed in the previous work.

New measurement techniques: In parallel, a major achievement of this work is the instrumental development, harnessing the benefits of quantum sensing in previously inaccessible material systems. The current work utilizes NV wide-field techniques to investigate mesoscopic magnetic patterns and spin fluctuations in tDT CrI₃. To our best knowledge, this is the first successful application of NV wide-field methods to emergent moiré superlattices. While we fully respect the significance of the pioneering work *Science* **374**, 1140 (2021), we would also like to point out that the technical improvement/development of the current study is clear and is not diminished.

In conclusion, this is a rapidly growing and advancing field. Taking advantage of state-of-the-art quantum sensing tools, our results enrich the current understanding of moiré magnetism physics and highlight the opportunities provided by NV centers in probing microscopic spin related phenomena on a two-dimensional flatland. We believe this work will generate interest amongst many branches on the forefront of quantum sensing and condensed matter physics research, and thus will be of great interest to the broad readership of *Nature Communications*.

Meanwhile, the authors take it for granted that structurally the twisted region of their sample is uniform. However, as generally known that, for small-angle twisted (0.3 degree) sample studied here, the superlattices can hardly be uniform in mesoscopic length scales. It's highly possible that the twisted region of the sample is already relaxed into random large domains with different and complicated interlayer stackings, which give rise to the domain structures observed by the wide-field imaging.

Response: We thank the reviewer for the excellent point. We agree with the reviewer that structural nonuniformity is expected in tDT CrI₃ and could be potentially relevant for their magnetic properties. Nevertheless, the proposed picture of extended domain phases in twisted CrI₃ remains valid as long as the magnetic domain size is larger than the local moiré period. Our NV measurement results (Figs. 2f and 2g) have explicitly shown that the lateral dimensions of the extended domain phases formed in twisted CrI₃ devices are in the micrometer range. Meanwhile, the MCD data (Fig. 2b), which is distinct from that of either 3L or 6L CrI₃, confirms the co-existence of ferromagnetic and antiferromagnetic phases due to the spatially modulated stacking geometries in tDT CrI₃ over the lateral dimensions ($\sim 1 \mu\text{m}$) of a beam spot. It is therefore safe to conclude that the local moiré period is smaller than the domain sizes in our tDT CrI₃ devices. Thus, we believe that the technical question raised by the reviewer, while very interesting, does not affect the validity and the main conclusion of our work.

Reviewers' Comments:

Reviewer #1:

Remarks to the Author:

The authors have addressed my questions. The revised manuscript has been improved. I would recommend the publication.

Reviewer #2:

Remarks to the Author:

Huang et al. have revised their manuscript in accordance with the proposed reviewer suggestions. All my questions have been adequately addressed. The combination of the authors changes and the additional discussion that has been added in order to provide context are beneficial to the overall readability and impact of the manuscript. I believe that the readership of Nature Communications will find this work to be thorough and interesting. I recommend publication.

Reviewer #3:

Remarks to the Author:

I'm still not fully convinced by the authors' argument. First, the uniformity of the sample is not guaranteed. It is just the target twist angle to be 0.3 degree. No other verifications of the final relaxed angle are provided. The real moiré wavelength is not known, and the spatial inhomogeneity can hardly be mapped. Hence the claim of "extended intrinsic moiré magnetic domains of opposite magnetizations appear over arrays of moiré supercells" is not valid. I suggest the authors to test a zero-degree sample (not the natural six-layer one) and see whether there is any domain structure observed. Second, the NV wide-field imaging measurement technique is not new as the authors claimed. It has been commonly used (ref.23-26 and newly added 39) including some of those by the same authors.

Response to reviews – NCOMMS-22-52510A

We thank the Reviewers 1 and 2 for their recommendation of our work. In this revision, we have addressed the Reviewer 3's additional comments. Below, we provide a point-by-point response.

Reviewer #1:

The authors have addressed my questions. The revised manuscript has been improved. I would recommend the publication.

Response: We thank the Reviewer for the recommendation of our manuscript.

Reviewer #2:

Huang et al. have revised their manuscript in accordance with the proposed reviewer suggestions. All my questions have been adequately addressed. The combination of the authors changes and the additional discussion that has been added in order to provide context are beneficial to the overall readability and impact of the manuscript. I believe that the readership of Nature Communications will find this work to be thorough and interesting. I recommend publication.

Response: We thank the Reviewer for the recommendation of our manuscript.

Reviewer #3:

I'm still not fully convinced by the authors' argument. First, the uniformity of the sample is not guaranteed. It is just the target twist angle to be 0.3 degree. No other verifications of the final relaxed angle are provided. The real moiré wavelength is not known, and the spatial inhomogeneity can hardly be mapped. Hence the claim of "extended intrinsic moiré magnetic domains of opposite magnetizations appear over arrays of moiré supercells" is not valid. I suggest the authors to test a zero-degree sample (not the natural six-layer one) and see whether there is any domain structure observed.

Response: We thank the reviewer for evaluating our manuscript again. We understand that sample quality is indeed a topic of discussion particularly for moiré devices. To further address the Reviewer's concern, we have performed selected area electron diffraction (SAED) and dark-field transmission electron microscopy (DF-TEM) measurements to characterize the crystallographic lattice structures of a prepared low-twist-angle tDT CrI₃ device as shown in Fig. R1. From SAED data presented in Fig. R1a, the local mean twist angle over a $\sim 900 \text{ nm} \times 900 \text{ nm}$ sample area was measured to be 0.4 ± 0.1 degree by fitting pairs third-order Bragg peaks (Fig. R1a), giving a moiré period of $\sim 100 \text{ nm}$. The experimentally measured twist angle shows a small variation (~ 0.1 degree) from the targeted value, within the accuracy of the stacking processes. A real-space DF-TEM

Figure R1. **a** SAED patterns of the third-order Bragg peaks of a low-twist-angle tDT CrI₃ device (red rectangles) from a surveyed area. **b** Averaged DF-TEM real space image constructed from $[\bar{3}300]$, $[\bar{3}030]$ and $[0\bar{3}30]$ Bragg peaks showing the characteristic hexagonal superlattice structure of the tDT CrI₃ device. The scale bar is 30 nm.

image presented in Fig. R1b features the superlattice structures with a periodicity that is $1/\sqrt{3}$ of the moiré wavelength, because the DF-TEM image is formed by the second-order superlattice peaks of twisted CrI_3 samples. Note that the observed distortion from the expected hexagonal lattice pattern is common for low-twist-angle tDT CrI_3 , which could be induced by lattice strain, relaxation, and local structural inhomogeneity.

The presented TEM measurements have explicitly shown the moiré patterns with a local moiré period smaller than the domain sizes in our tDT CrI_3 devices, supporting the proposed picture of extended domain phases in twisted CrI_3 (please find more detailed information on this point in our previous round response). Last, we comment that an absolute “zero-degree” sample suggested by the reviewer is technically challenging to realize due to the uncertainty ($\sim\pm 0.15$ degree) of twist angle in the real experiments. So, here we employ SAED and DF-TEM to further clarify the microscopic structural details of the prepared low-twist-angle tDT CrI_3 devices. In this revision, Figure R1 and the relevant discussions have been included in the revised Supplementary Information.

Second, the NV wide-field imaging measurement technique is not new as the authors claimed. It has been commonly used (ref.23-26 and newly added 39) including some of those by the same authors.

Response: While we agree with the Reviewer that the presented experimental techniques have been previously used to investigate many interesting condensed matter systems, here we also would like to highlight that the current work is the first successful demonstration of NV wide-field measurements of emergent magnetic moiré superlattices (to our best knowledge). The immediate scientific impact of our study is clear, and we share the optimism that the presented results will stimulate more research interests and efforts in this rapidly growing and advancing field.

Reviewers' Comments:

Reviewer #3:

Remarks to the Author:

In the revised manuscript, the authors performed updated SAED and DF-TEM measurements, from which distorted and mosaic-like domain patterns of the tDT CrI3 can be observed. The structure is clearly much more complicated than the simplified moiré pattern drawn in the schematic Fig.1c and I presume the measurement is from a different sample. These points should be made more transparent to the readers and would help to justify the validity of the study. Overall, I appreciate the authors' efforts in measuring the microscopic lattice structure and would recommend publication.

Response to reviews – NCOMMS-22-52510B

Reviewer #3:

In the revised manuscript, the authors performed updated SAED and DF-TEM measurements, from which distorted and mosaic-like domain patterns of the tDT CrI₃ can be observed. The structure is clearly much more complicated than the simplified moiré pattern drawn in the schematic Fig.1c and I presume the measurement is from a different sample. These points should be made more transparent to the readers and would help to justify the validity of the study. Overall, I appreciate the authors' efforts in measuring the microscopic lattice structure and would recommend publication.

Response: We thank the Reviewer for devoting time to evaluate our manuscript again. We are very happy to hear the reviewer's recommendation of our work. Reviewer's insightful suggestions help us further improve the manuscript and strengthen our conclusions. In this revision, we have clearly stated that the microscopic lattice structure of prepared low-twist-angle tDT CrI₃ devices typically shows distortions from the expected hexagonal superlattice pattern in the first paragraph on page 3 as follows:

“...It is worth mentioning that the microscopic lattice structure of prepared low-twist-angle tDT CrI₃ typically shows certain distortions from the expected hexagonal superlattice pattern, which could be induced by lattice strain, relaxation, and local inhomogeneities.....”